# Ethylene Scavenging Films Based on Ecofriendly Plastic Materials and Nano-TiO_2_: Preparation, Characterization, and In Vivo Evaluation

**DOI:** 10.3390/polym16060853

**Published:** 2024-03-20

**Authors:** Alba Maldonado, Tomas Aguilar, Carolin Hauser, Gerd Wehnert, Dominik Söthje, Herbert Schlachter, Alejandra Torres, Julio Bruna, Ximena Valenzuela, Francisco Rodríguez-Mercado

**Affiliations:** 1Packaging Innovation Center (LABEN–Chile), Universidad de Santiago de Chile, Obispo Umaña 050, Santiago 9170201, Chile; tomas.aguilar@usach.cl (T.A.); alejandra.torresm@usach.cl (A.T.); julio.bruna@usach.cl (J.B.); ximena.valenzuela@usach.cl (X.V.); 2Center for the Development of Nanoscience and Nanotechnology (CEDENNA), Universidad de Santiago de Chile, Alameda 3363, Santiago 9170022, Chile; 3Department of Applied Chemistry, Nuremberg Institute of Technology Georg Simon Ohm, Keßlerplatz 12, 90489 Nuremberg, Germany; carolin.hauser@th-nuernberg.de (C.H.); gerd.wehnert@th-nuernberg.de (G.W.); dominik.soethje@th-nuernberg.de (D.S.); herbert.schlachter@th-nuernberg.de (H.S.); 4Department of Food Science and Technology, Faculty of Technology, Universidad de Santiago de Chile, Avenida Víctor Jara 3769, Santiago 9170124, Chile

**Keywords:** ethylene removers, titanium dioxide, eco-friendly films, shelf life

## Abstract

It is known that ethylene plays an important role in the quality characteristics of fruits, especially in storage. To avoid the deterioration of fruits caused by ethylene, titanium dioxide (TiO_2_) has been used due to its photocatalytic capacity. The aim of this study was to develop films based on two types of biopolymers, Mater-Bi (MB) and poly-lactic acid (PLA), with nanoparticles of TiO_2_ and to determine their ethylene removal capacity and its application in bananas. First, the films were fabricated through an extrusion process with two different concentrations of TiO_2_ (5 and 10% *w/w*). Then, the films were characterized by their structural (FTIR), morphological (SEM), thermal (DSC and TGA), dynamic (DMA), barrier, and mechanical properties. The ethylene removal capacities of the samples were determined via gas chromatography and an in vivo study was also conducted with bananas for 10 days of storage. Regarding the characterization of the films, it was possible to determine that there was a higher interaction between PLA with nano-TiO_2_ than MB; moreover, TiO_2_ does not agglomerate and has a larger contact surface in PLA films. Because of this, a higher ethylene removal was also shown by PLA, especially with 5% TiO_2_. The in vivo study also showed that the 5% TiO_2_ films maintained their quality characteristics during the days in storage. For these reasons, it is possible to conclude that the films have the capacity to remove ethylene. Therefore, the development of TiO_2_ films is an excellent alternative for the preservation of fresh fruits.

## 1. Introduction

Ethylene (C_2_H_4_) is a gaseous plant hormone that exhibits both beneficial and detrimental effects in the postharvest storage of fruits and vegetables (F&Vs). Beneficial effects include the development of characteristic colors, tastes, textures, and flavors. On the other hand, it can have detrimental effects, such as promoting senescence, resulting in discoloration, softening, and increased susceptibility to spoilage, all of which reduce the product’s shelf life. F&Vs release different amounts of ethylene gas, and their susceptibility to ethylene exposure is also different [1]. For example, in climacteric fruit, ethylene accelerates ripening, causing excessive fruit softening, color and texture changes, modifications in sugar content, and synthesis of volatile aroma compounds. Meanwhile, in non-climacteric fruit, ethylene stimulates senescence, often characterized by the yellowing of green tissues due to promoting chlorophyll degradation, as well as accelerating the fruit’s toughening and wilting [2]. The overripening of fruits leads to excessive softening, resulting in spoilage and damage during shipping and handling. Therefore, slowing the process of ripening and senescence can extend the storage and shelf life of fresh fruits. Increasing the shelf life of fruits not only saves growers postharvest losses but also benefits consumers by providing longer-lasting freshness and greater value in the fruit [3]. Considering the adverse effects of ethylene on fruits and vegetables, various strategies have been developed with the goal of reducing or inhibiting its production and negative impacts.

Research aimed at removing ethylene from storage atmospheres for various types of produce has primarily focused on developing packages that either adsorb or actively scavenge ethylene. Active packaging is used to remove unwanted ethylene from the headspace of a package via absorption, adsorption, or scavenging. This is typically achieved by incorporating a physical or chemical adsorbent into the packaging material or adding it to the package as a sachet [4]. Potassium permanganate has been widely utilized to regulate ethylene concentrations in packaging due to its cost-effectiveness and convenience. However, the toxic potential of potassium permanganate to the human body has been reported [1,5]. Photo-catalysis presents a promising method for preventing fruit ripening at room temperature while consuming less energy and incurring lower costs. TiO_2_ nanoparticles have been proposed as a favored solution for ethylene removal within these systems. TiO_2_ has been demonstrated to be highly appropriate for various environmental and energy applications due to its suitable valence and conduction band positions, long-term stability, non-toxic nature, cost-effectiveness, and high oxidizing power. Moreover, the US Food and Drug Administration (FDA) has granted approval for using TiO_2_ in the food industry [6]. This semiconductor has attracted increasing attention owing to its potential application in the degradation of organic contaminants. In nature, TiO_2_ has three polymorphs, namely, anatase, rutile, and brookite. Rutile is the stable phase, while anatase and brookite are metastable phases [7]. It is generally accepted that anatase displays much higher photocatalytic activities than both rutile and brookite [8,9]. The mechanism of scavenging is through reactive oxygen species, which are produced on the TiO_2_ surface following exposure to UV light (<380 nm), thus further oxidizing ethylene into carbon dioxide and water. This catalytic system does not need high temperatures or pressure for scavenging, and it is less energy-demanding compared to thermal catalytic oxidation [10]. The above properties are what make this catalytic system an excellent option for the removal of ethylene from the transport and sale conditions of F&Vs sensitive to this gas.

In recent years, several studies have reported the development of titanium dioxide (TiO_2_)-based ethylene scavenging materials and their effects on the shelf lives of various fruits. To evaluate ethylene scavenger properties, several nanocomposite films of chitosan nano-TiO_2_ have been prepared by solution casting (0, 0.25, 0.50, 1.00 y 2.00 wt.%) [11]. Furthermore, the impact of UV radiation has been evaluated at 25 °C and 85% RH (relative humidity). A systematic decay of ethylene has been observed as a function of the TiO_2_ concentration in the films. Interestingly, there was no significant difference in the ethylene removal capacities between films with 1 and 2 wt.% TiO_2_, which can be attributed to the formation of TiO_2_ agglomerates over 1 wt.%. An in vivo evaluation has also been performed on cherry tomatoes [12]. The tomatoes were packed in bags made from chitosan nanocomposite films with 1 wt.% TiO_2_. The UV-treated active film effectively preserved the quality of the tomatoes, as evidenced by their firmness, color, lycopene content, and soluble solid content. To improve the dispersion of nano-TiO_2_ (anatase) within an LDPE matrix, a study used a precursor material made from LDPE and TiO_2_ masterbatch to produce active LDPE films containing 1 wt.% of TiO_2_ [13]. The resulting active LDPE films were then used to produce packaging for fresh strawberries, which were stored at 4 °C in the darkness for 14 days. The active system exhibited a capacity to maintain higher organoleptic and nutritional qualities compared to that of an ethylene control package. Recently, there have been reports on the use of nanofibers based on nano-TiO_2_ as an ethylene scavenger. Böhmer-Maas et al. [14] reported the fabrication of sachets by deposition of zein/TiO_2_ nanofillers on aluminum plates. These sachets (15 × 7 cm) were then stored inside an airtight container (750 mL) containing cherry tomatoes, and this system was stored at 25 °C. The monitoring of ethylene inside the container did not show any significant changes compared to the control during the 22 days of the experiment. In contrast, the photocatalytic activities of films based on polyacrylonitrile (PAN)/TiO_2_ nanofiber were confirmed against a banana fruit ripening. In this study, and according to a visual evaluation, it was possible to detect a delay in the ripening of bananas in contact with the active fibers [15]. Alternatively, TiO_2_ nanoparticles were incorporated into biodegradable poly (butylene adipate-co-terephthalate) (PBAT) and thermoplastic cassava starch (TPS)-blended films to produce nanocomposite packaging via blown-film extrusion [16]. Banana fruit packaged in nanocomposite films recorded a slower darkening color change and enhanced shelf life with an increasing TiO_2_ content from 1 to 5 wt.%.

This study aimed to evaluate the effect of nano-TiO_2_ on the physicochemical properties of PLA and MB, as well as the ethylene scavenging capacity induced by UV and Vis light. Furthermore, the most effective materials were tested through in vivo assays using banana as an ethylene-sensitive fruit.

## 2. Materials and Methods

### 2.1. Materials

To elaborate different films, nanoparticles of titanium dioxide (nano-TiO_2_) anatase (Sigma Aldrich, 21 nm, 99.5%, San Luis, MO, USA), Mater-Bi^®^ EF51L (Novamont SpA, Milan, Italy) and PLA Ingeo Biopolymer 2003D (NatureWork, Plymouth, MN, USA) were used. For the in vivo evaluation, bananas (Cavendish, Ecuador) were provided by the “Anish” Fruit Distribution Company (Santiago, Chile).

### 2.2. Fabrication of Active Films

To produce the nanocomposite films, the polymeric matrices PLA and Mater-Bi^®^ (MB) were used with different contents of nano-TiO_2_ (0, 5, and 10 wt.%). The nanocomposite films were produced by extrusion using a 20 mm corotating laboratory twin-screw extruder LTE20 (Labtech Engineering Co. Ltd., Samut Prakan, Thailand). The extrusion conditions included a screw feed speed of 15 rpm, a twin-screw speed of 30 rpm, a die temperature of 190 °C, and a temperature profile of 160–185 °C and 120–180 °C for PLA and MB, respectively. The films were collected in a chill-roll attachment Labtech LBCR-150 (Samutprakarn, Thailand) at 3 rpm.

### 2.3. Characterization of Active Films

#### 2.3.1. Morphological Properties

To determine the principal functional groups of films, a Fourier transform spectroscopy (FTIR) analysis was carried out in an IR-ATR spectrometer (Bruker, Alpha, Germany) with a wavelength range from 4000 cm^−1^ to 400 cm^−1^. The absorbance was measured using 24 scans and a resolution of 2 cm^−1^.

Scanning electron microscopy (SEM) images were obtained on a microscope (Tescan, Vega, Brno, Czech Republic) from 3 to 20 kV. The samples were prepared with a gold–palladium film coated on a Hummer 6.2 metallizer.

#### 2.3.2. Thermal Properties

Differential scanning calorimetry (DSC) analyses were carried out in a Mettler Toledo DSC-822e calorimeter (Bern, Switzerland). Samples (≈5 mg) were sealed in 40 μL aluminum crucibles and heated from 30 °C to 200 °C at a rate of 10 °C/min under a dry nitrogen purge. The results obtained correspond to the second heat, as the first heat was performed to delete the thermal history of the material. The crystallinity of the films was calculated according to Equation (1).
(1)$$x_{c}\left(\%\right)=\frac{\left(∆H_{m}-∆H_{cc}\right)}{∆H_{0}}\times 100$$
where $x_{c}\left(\%\right)$ is the crystallinity of PLA in the films; $∆H_{m}$ represents the heat of melting; $∆H_{cc}$ is the value of the cold heat of crystallization; $∆H_{0}$ (97.3 J·g^−1^) is the theoretical melting enthalpy of 100% crystalline PLA. In the case of MB, as this is a commercial mixture of polymers, this value is unknown. Then, $x_{c}\left(\%\right)$ values were not reported for MB nanocomposite films.

Thermogravimetric analyses (TGA) of films were determined with a TGA/DSC 1 analyzer (Mettler Toledo, STARe System, Bern, Switzerland) using the STARe V.12.0 software (Bern, Switzerland). In an alumina crucible, 10 mg of the sample was weighed (70 µL approximately). The experiment was conducted in a high-purity dynamic nitrogen flow of 50 mL/min at a heating rate of 10 °C/min. The analysis was performed in a temperature range of 25 to 500 °C.

#### 2.3.3. Dynamics Properties

A dynamic mechanical analyzer (DMA) DMA 1 instrument (Mettler Toledo, STARe System, Bern, Switzerland) was used to evaluate the effect of the nano-TiO_2_ content on the viscoelastic properties of the different nanocomposites. The analysis was carried out in the bending mode at a frequency of 1 Hz on test bars (40 × 6 mm) cut from compression-molded sheets. The analysis was performed from 30 °C to 70 °C using a heating rate of 10 °C/min.

#### 2.3.4. Mechanical Properties

The Young modulus (YM), tensile strength (TS), and elongation at break (EB) of the films were measured at room temperature on a universal testing machine (Zwick/Roell, Z005, Ulm, Germany) according to the standard of [17]. For this analysis, specimens were cut from the samples into sizes of 210 mm × 25 mm. These samples were previously conditioned at 25 °C and 50% relative humidity (RH) for 48 h. The specimens were subjected to tensile tests using a distance between the grips of 125 and 50 mm for PLA and MB, respectively. A test speed of 100 mm/min and a 0.1 N preload were selected. The YM, TS, and EB were obtained with TestXpert v1102 software (Zwick/Roell, Z005, Ulm, Germany).

### 2.4. Ethylene Removal Study

The ethylene removal kinetics of nano-TiO_2_ films were carried out according to the methodology described below. First, pieces of film (15 × 60 mm) were introduced into a 22 mL vial with 100 μL of distilled water inside. Then, the vials were sealed using a silicone/PTFE septum and an aluminum hole cap. Then, an ethylene–nitrogen mix gas (12 ppm) was injected into each of the vials for 1 min; a rotameter was used to ensure the flow rate, and after that, the vials were placed inside a special ultraviolet-UV or visible-VIS light chamber and were irradiated for 5 min at 20 °C. Then, samples from the headspace were analyzed by gas chromatography for 7 days. The ethylene quantification was conducted in a Perkin Elmer Clarus 580 gas chromatograph (Waltham, MA, USA), with a Head Space Turbo Matrix 40-Perkin Elmer sampler (Waltham, MA, USA) and an RtTM-alumina PLOT column of a 50 m length and 0.53 mm diameter. The gas chromatograph conditions were the same as those used in previous work [18]. The ethylene removal capacity of different samples was calculated according to Equation (2):(2)$$Final\;C_{2}H_{4}\;removal\left(\%\right)=100-\left(\frac{C_{2}H_{4\;final}}{C_{2}H_{4\;initial}}\times 100\right)$$
where $C_{2}H_{4\;final}$ is the ethylene concentration at day 7, and $C_{2}H_{4\;initial}$ is the ethylene concentration at day 0.

### 2.5. In Vivo Evaluation in Banana

The banana was placed inside a 25 × 10 cm bag, which was made from different films. The bag had only 3 sealing zones, leaving one open to allow the fruit to breathe. The packed bananas were kept inside a chamber at 14 °C with a RH of 70% for 10 days. Every two days, the fruits were characterized. This characterization included an analysis of the maturity index using a standardized color chart for bananas according to the Von Loesecke scale [19], pulp texture with a texturometer (Zwick/Roell, Z005, Ulm, Germany), peel color with a colorimeter (Konica Minolta, CR 400, Tokyo, Japan), soluble solids of the pulp through a refractometer (Atago, Master, Saitama, Japan) and a pH evaluation through a pH-meter (Hanna, Smithfield, RI, USA). Additionally, the starch index was measured through the Lugol staining test (5 g of I_2_ and 10 g of KI in 85 mL of distilled water).

### 2.6. Statistical Analysis

Each analysis was performed in triplicate, and the data were analyzed by ANOVA and Tukey’s multiple range test using Statgraphics Centurion XV.II^®^ (Statgraphics Technologies Inc., The Plains, VA, USA). The statistical significance was determined at a level of *p* < 0.05.

## 3. Results

### 3.1. Characterization of Active Films

To determine the effect of the addition of different concentrations of nano-TiO_2_ into the films, morphological, thermal, mechanical, dynamic, barriers, and optical properties were measured.

#### 3.1.1. Morphological Properties

FTIR spectroscopy allows characterizing the functional groups of polymers through the identification of their absorption bands [20]; therefore, it can evidence interactions between the different components present in the films and changes produced by the addition of active components such as nano-TiO_2_. Regarding the films elaborated with PLA, Figure 1A displays the FTIR spectra. All samples show a similar spectrum with the characteristic absorption bands of PLA, which have been previously reported by [20]. Thus, it is possible to distinguish the vibrational peaks between 2993 and 2941 cm^−1^ corresponding to the stretching vibration of the methyl (-CH_3_) and methylene (-CH_2_-), respectively. The band at 1757 cm^−1^ represents the stretching of carbonyl C=O in the ester group, and the bands at 1455, 1378, and 1358 cm^−1^ can be attributed to the symmetrical and asymmetrical stretching deformation vibrations of the aliphatic groups -CH_2_- and -CH_3_. The absorption bands between 1242 and 1048 cm^−1^ are characteristic of the C-O stretching vibration and the deformation of the O-CH-CH_3_ in the ester group, respectively [21]. In addition, a band between 500 and 900 cm^−1^ might be associated with the deformation vibrational modes of the Ti-O-Ti bonds. The peak corresponding to the Ti-O-Ti bonds with a stretching mode of vibration is observed at 452 cm^−1^ in the samples containing nano-TiO_2_ (PLA 5 and PLA 10%) [22]. This indicates the presence of the nanoparticle and its interaction with the material.

On the other hand, and with reference to the nanocomposites fabricated with MB, the assignment of bands was carried out taking as a reference a previous study where the presence of PLA, poly (butylene adipate-co-terephthalate (PBAT) and thermoplastic starch (TPS) was evidenced [23]. The FTIR spectra of MB/nano-TiO_2_ nanocomposites are shown in Figure 1B. It can be observed that the MB control spectra show the PLA in its composition. For example, the bands between 2800 and 2900 cm^−1^ correspond to the stretching of the axial -CH_3_ groups present in aliphatic groups, and the bands between 1700–1250 cm^−1^ correspond to the carbonyl groups (C=O) and to the C-O bonds attributed to esters [23]. On the other hand, the signals near 1460 cm^−1^ correspond to a phenyl group stretching absorption band of the aromatic ring from PBAT [23], which is also in accordance with the bands between 900 and 400 cm^−1^ that could correspond to the bending of benzene substituents [24]. Likewise, the bands found at 1250 cm^−1^ correspond to the stretching of the C-O-C group of the glucose ring, while the bands at 1086 cm^−1^ and 1134 cm^−1^ correspond to the stretching of C-O in the C-O-H group, supporting the presence of starch [25]. The presence of the active ingredient within the matrix is indicated by the peak at 452 cm^−1^ in the samples containing nano-TiO_2_, as discussed previously.

The morphology of the samples was analyzed by SEM microscopy. Figure 2 shows the photographs corresponding to each film, where the white particles indicated with arrows correspond to nano-TiO_2_. As can be seen, the PLA films show a good distribution of nano-TiO_2_ within the film. Although it is possible to observe agglomerations of the active compound in certain areas of the material, their sizes are small. This could indicate that there is a good interaction between the active substance and PLA. Such behavior would promote the generation of a larger contact area for the active substance, which is a favorable condition for the photocatalytic action of TiO_2_. In contrast, the films made with MB exhibit a different behavior, where higher levels of nano-TiO_2_ agglomerates can be observed, consistent with the content of the inorganic compound (Figure 2I). This could indicate that the interaction of MB with TiO_2_ is lower than that of the PLA matrix.

#### 3.1.2. Thermal Properties

The thermal properties of the nanocomposite films were determined through DSC and TGA analyses, and the results are presented in Table 1 and Table 2. Regarding the PLA films, the glass transition temperature (Tg) is around 58 °C for all samples, and there are no differences between them (*p* > 0.05). In addition, the cooling crystallization temperature (Tcc) and melting temperature (Tm) are similar for both the PLA and PLA 10%. However, the PLA 5% sample shows a significant decrease in Tcc. This result would be evidencing a nucleating effect of the nanoparticles. Although there is evidence of agglomerate formation in this nanocomposite film, this fact would indicate that isolated nanoparticles would also exist. The crystallization percentages (X%) of the PLA and PLA 5% samples do not show significant differences (*p* > 0.05); nevertheless, PLA 10% decreases with respect to the control. The behavior could be attributed to the TiO_2_ agglomeration, which can cause an over-saturation of the nucleation sites in the PLA [26]. All these changes in the thermal properties of PLA films might indicate there are no movements in the polymer chains due to the addition of nanoparticles, and it could be suspected that there is an interaction between the PLA carbonyl groups and hydroxyl groups on the surface of the nanoparticles [27].

On the other hand, the MB samples show a different behavior than PLA. Table 2 summarizes the thermal properties of the MB nanocomposite samples, which can be divided into three main zones. The first zone is around 60 °C and it can be attributed to the T_g_ of PLA. After this zone, an endothermic peak can be observed between 122 and 127 °C, which is associated with the melting temperature (T_m_) of PBAT [28]. Finally, endothermic peaks ranging from 164 to 170 °C can be observed, which can be related to the melting of PLA [29] and TPS [30]. Regarding the TPS, it has been reported that this material has a wide melting zone (160 and 380 °C), which is dependent on the amylose/amylopectin ratio and the content and structure of the plasticizer used [30].

The results of the thermogravimetric analysis (TGA) are summarized in Figure 3. The PLA nanocomposites (Figure 3A,B) show similar thermograms and the corresponding DTG curves. However, at the end of the degradation process, it can be appreciated that the PLA control decomposes completely, leaving a low residue content (~0.3%). In contrast, PLA 5% and PLA 10% leave solid residues, which is in accordance with the concentration of nano-TiO_2_ in the nanocomposites (~4.2 and ~9.5%)

The degradation of the PLA, PLA 5%, and PLA 10% film samples start at 351.7 °C, 352.4 °C, and 355.4 °C, respectively, with a maximum degradation temperature close to 369 °C. On the other hand, it can be observed that the solid residues for the MB samples are 5.7, 6.0, and 14%. Unlike the PLA samples, three degradation zones can be observed, which would correspond to the degradation temperatures of the different components from MB. The first peak observed would correspond to TPS, whose degradation temperature starts at 320 °C. The second peak could be attributed to PLA, whose degradation temperature ranges from 340 to 380 °C, and the third peak could correspond to PBT, whose degradations are 395 to 410 °C. The MB 10% sample shows a degradation process at 358 °C, corresponding to the degradation of TPS. This behavior could indicate that the addition of nano-TiO_2_ in polymeric matrices improves their thermal properties and acts as a thermal barrier in the early stages of decomposition [31]. In addition, this could be explained by possible interactions between starch with nano-TiO_2_, forming hydrogen bonds with the -OH groups of starch [16].

#### 3.1.3. Dynamics Properties

This analysis was conducted to evaluate the compatibility of active TiO_2_ with different polymeric matrices, i.e., the interfacial interactions between both phases. The DMA study on the viscoelastic behavior of polymeric materials comprises three analyses: first, the storage modulus (E′) represents the energy stored in the elastic region by the polymeric material; second, the loss modulus (E″) represents the dissipation of energy from the polymer in the form of heat; and third, the ratio of the storage modulus to the loss modulus, represented by tan delta (tan δ), also known as the damping factor, describes the material’s resistance to deformation [31]. Figure 4 indicates the dependence of E′ and tan δ of different films.

Regarding the storage modulus (E′), all samples exhibited similar behavior with an increasing temperature, i.e., E′ decreases as the temperature increases. The E′ curves can be divided into three stages. The first stage occurs below the glass transition region and can be referred to as the glassy region. Here, the movement of polymer chains is restricted due to the low mobility of the frozen and packed molecule arrangement, resulting in a higher storage modulus. In the second stage, the glass transition region occurs when the arrangement of packed molecules begins to collapse, leading to the increased molecular mobility of polymer chains and a drastic reduction in the storage modulus. Finally, in the third stage, a rubbery region is reached, characterized by a second plateau. [32]. For the PLA samples (Figure 4A), it was observed that the storage modulus exhibits a long plateau in the temperature range from 30 to 55 °C for PLA and PLA 5%, while for PLA 10%, this plateau is shorter (30 to 45 °C). Then, a drastic drop from 55 to 60 °C was observed for PLA and PLA 5%, and from 45 to 50 °C for PLA 10%. Whereas the MB samples (Figure 3B) show a shorter plateau than PLA (30 to 40 °C); the modulus drop is not drastic but decreases continuously between 40 and 60 °C. When comparing the dynamic behavior between PLA and MB PLA films at room temperature (25 °C), the PLA films exhibit much higher E′ values (2500–3500 MPa) compared to the MB films (200–300 MPa).) It is well known that when the E′ value is higher or increases, it indicates that there is a better interaction between the matrix and the nanoparticles. As a result, it could be said that there is a better interfacial interaction and a good dispersion of nano-TiO_2_ in PLA, which is coherent and related to the morphological properties previously discussed.

On the other hand, the variation in the damping factor tan δ as a function of the temperature is shown in Figure 4B,D. In these graphs, the maximum point of the peak corresponds to the glass transition temperature (Tg). Thus, the Tg of the PLA, PLA 5%, and PLA 10% films is 65.0 ± 1.5, 62.8 ± 0.5, and 54.7 ± 1.4 °C, respectively, and for the MB, MB 5%, and MB 10% samples, it is 59.7 ± 0.7, 54.0 ± 1.4, and 59.0 ± 1.1 °C, respectively. Additionally, the intensity of tan δ indicates the mobility of the polymer chains at a given temperature and accounts for the way in which the materials absorb or disperse the energy during deformation [33], and when the intensity of tan δ is higher, it means that the energy is dispersed and that the sample has a viscous behavior. On the contrary, if tan δ is lower, the energy is stored, exhibiting an elastic behavior [34]. In this case, it could be said that the PLA samples have a viscous behavior while MB is elastic; these results are in agreement with the mechanical properties, which will be discussed in the next section.

#### 3.1.4. Mechanical Properties

Mechanical properties are very important in the design of food packaging, especially when it is based on eco-friendly materials. Thus, to determine the effect of nano-TiO_2_ on the mechanical properties of PLA and MB films, Young’s modulus (YM), as well as the tensile strength (TS) and elongation at break (EB), were determined (Table 3).

In general, it is observed that there are significant differences (*p* < 0.05) between the PLA and MB samples with different nano-TiO_2_ contents. Table 3 also shows the thickness of the films, and it is possible to observe that there are differences between the samples, where the MB films have a higher thickness than the PLA films; moreover, the addition of nano-TiO_2_ affects this parameter, especially in the PLA samples; where PLA 10% has the lower thickness (14.10 ± 0.8 µm), and it is related to the behavior of mechanical properties. The reduction in thickness for the PLA sample with 10% TiO_2_ can be explained by the change in viscosity of the molten material caused by the interactions between the nanoparticles with the polymer chains. It is known that nanoparticles can drastically reduce the viscosity of a polymer in the molten state and that this effect can be associated with polymer degradation or a lubricating effect of the nanoparticles, which improves the mobility and flow of the polymer chains in the melt [35,36]. The change in the rheological properties of the melt has been reported in other studies, where it is recognized that TiO_2_ has the tendency to absorb polymeric chains on its surface, thus creating a free volume around the nanoparticles that affect the viscosity of the melt [36]. In this work, polymer degradation would be discarded based on the DSC analysis, which did not show a change in the Tg value with TiO_2_ content, which would suggest a degradation of the polymer due to a decrease in its molecular weight. Thus, the explanation would be in agreement with a change in the viscosity of the melt. This fact is consistent with what was observed when collecting the material from the extruder head to the chill-roll system. In the case of the PLA with 10% nanoparticles, the viscosity of the melt was significantly lower than that of the other materials. This was evident from a noticeable change in the appearance of the material and made it difficult to transfer it from the extruder head to the chill-roll system. This would be consistent with the other results obtained in this study, where the coexistence of nanoparticles and their aggregation within the PLA matrix is evidenced.

Regarding the PLA films, there was evidence that PLA 0% and PLA 5% do not show statistically significant differences (*p* > 0.05) for the YM and TS parameters; however, the PLA 10% films show important differences with those films. In that sample, all physical parameters are lowest, which agrees with a high sensitivity to breakage compared to the other films. In addition, this behavior will also be affected by the film’s thickness, where PLA 10% is the lowest thickness film, as mentioned before. Despite MB films having similar thicknesses, these samples show a decrease in the YM, TS, and EB parameters according to the increase in nano-TiO_2_ content. In comparison with the PLA films, the MB films show lower YM and TS values and higher values of EB. These results can be associated with the composition of MB.

According to the previous results, it has been evidenced that MB is a mixture of different polymeric matrixes, such as PLA, PBAT, and TPS. The presence of TPS in this commercial material would confer it with more elastic characteristics and much higher breaking strength compared to the PLA. Based on these results, it could be indicated that the addition of nano-TiO_2_ improves the mechanical properties up to a concentration of 5% due to good interfacial adhesion between the matrix and the active. This is justified by an increase in the strength of the TiO_2_ particles, reducing the deformability of the polymer molecules. In addition, the incorporation of nano-TiO_2_ reduces the movement of the polymeric structures, which improves the stiffness of the material [37]. This behavior would be important for PLA films; however, at a higher concentration (10%) of TiO_2_, the mechanical properties are affected, and this could be explained by the excessive agglomeration of the inorganic active and, in the specific case of MB, a low adhesion of the active with this matrix. These results are in agreement with the analyses of the morphology and dynamic properties that show a lower interaction between nano-TiO_2_ and MB.

### 3.2. Ethylene Removal Study

To comprehend the ethylene removal capacity of the films, the evolution of the ethylene concentration was determined for 7 days. This analysis also evaluated the incidence of the type of irradiation (UV light and VIS light) on the ethylene remotion capacity of the different films (Figure 5).

In general, it can be observed that the ethylene concentration decreases continuously as the days progress. The control samples (0%) show a faster decrease in the olefin concentration, but then it stabilizes (~day 3), while the samples with nano-TiO_2_ take a little longer; although, at the end of the kinetics, the samples with 5% and 10% TiO_2_ present better results. In addition, differences can be observed in terms of the type of irradiation provided to each film, where greater removal activity is observed in the samples irradiated with UV light (Figure 5A). Moreover, the PLA presents a higher and faster removal activity with respect to the MB. It is important to mention that the kinetic behavior of the samples does not have a defined pattern since it is very variable for each of the conditions studied. Finally, Table 4 presents the percentage of ethylene degradation on day 7 by different conditions.

Based on these results, it could be indicated that there are three mechanisms for ethylene removal, which are explained below: (i) There is ethylene physisorption by the material, where the aliphatic groups of PLA and MB interact with ethylene; (ii) it is the sorption of ethylene on the TiO_2_ surface that remains within the matrix but is not activated; and (iii) the photocatalytic action of TiO_2_, which develops on the surface of the material and is activated to a greater degree by ultraviolet light, is related to the thickness of the film.

According to the irradiation type factor, the samples irradiated with UV light show higher C_2_H_4_ removal with respect to VIS light. These results were expected since UV irradiation has enough energy to activate nano-TiO_2_ [38]; however, the samples irradiated with VIS light also remove C_2_H_4_, although in smaller quantities. The activity, in this case, can be explained since VIS light has 5% UV light in its composition [39]. It is also possible that TiO_2_ is also activated under this type of irradiation, along with the other factors discussed above.

PLA is the matrix that presents a higher percentage of C_2_H_4_ removal compared to MB. This could be related to the higher interaction of nano-TiO_2_ with PLA than MB, as previously observed by SEM microscopy and the DMA analysis. In Table 4, it can be observed that the highest removal values correspond to the samples with 5% of nano-TiO_2_. Consistent with earlier observations, a higher concentration of the active ingredient favors the agglomeration of the inorganic nanoparticles, which decreases the photocatalytic activity due to the reduced contact area of the active compound. Based on these results, it was possible to conclude that the PLA 5% sample irradiated with UV light achieved a higher final ethylene degradation, reaching a value of 68%, compared to the other samples. For this reason, it was decided to use this film for the development of the in vivo study.

### 3.3. In Vivo Study in Banana

To determine the functionality of the elaborated films, an in vivo study was carried out on bananas. Physiological parameters of both the peel (ripeness and color) and the fruit pulp (texture, soluble solids (SS), pH, and starch index) were evaluated. The films used for this study were PLA and PLA 5%, with additional control samples consisting of bananas without film contact.

#### 3.3.1. Ripeness Index and Color of the Peel

The evolution of the ripeness index and the peel color of the fruit in contact with the PLA films were evaluated during the 10 days of storage. Table 5 shows the color parameters (L*, a*, b*, and ΔE) of the fruit peel. As can be observed, the L* parameter, which is indicative of fruit brightness, decreases over time in all samples, showing no significant differences among them. Parameter a*, which indicates a tendency to red color when it is positive or a tendency to green when it is negative, increases according to the evolution of time, showing significant differences between samples. It is important to mention that the bananas packaged with PLA 5% retain a tendency towards remaining green until day 10, while the other samples indicate a change in this parameter, which is much more evident in the control sample. On the other hand, parameter b* decreases over time, maintaining positive values that show a tendency towards yellowing in the fruits, with no significant differences between the samples. This is consistent with the increase in opacity of the banana peel due to the reduction in metabolic activity that occurs when this fruit is stored at temperatures below 25 °C [40]. Finally, the ∆E value shows that there is a lower color difference between the control and the bananas in contact with PLA 5%.

#### 3.3.2. Fruit Pulp Characterization

The evolution of both the texture and soluble solids of fruit pulp are represented in Figure 6. Regarding texture (Figure 6A), it can be observed that there are significant differences between the samples and days of evaluation. It is evident that the control sample, as time progresses, loses texture to a greater degree than the PLA 0% and PLA 5% samples. At the end of the test, the PLA 5% sample exhibited a higher texture value (~30 N) compared to the other samples. This result would suggest the effect of nano-TiO_2_ on reducing endogenous ethylene around the fruit, thereby potentially delaying fruit ripening. On the other hand, the results of the evolution of the soluble solids (SS) content are depicted in Figure 6B. Here, the control sample shows a significant increase in SS during 10 days in comparison to the samples packed in PLA 0% and PLA 5%. Additionally, at day 10, the bananas packed in PLA 5% present the lowest SS value (~17° Brix). This result is consistent with the reasoning used to explain the texture results.

Finally, Figure 7 represents the evolution of the coloration generated by the Lugol with the starch from banana samples. These results show a decrease in the coloration generated over time for all samples. In addition, the fruits packed with PLA and PLA 5% show a higher starch conservation with respect to the control at the end of the measurement. These results are related to the SS content, which is higher over time due to the transformation of starch into simpler sugars, as mentioned above. It is well known that starch degrades, as do other structures such as chlorophyll and cell walls, allowing fruit flesh to soften and lose texture as it ripens.

## 4. Conclusions

Through the extrusion process, it was possible to elaborate active films from the incorporation of nano-TiO_2_ in eco-friendly materials such as PLA and MB. According to the results of the characterization of these films, it was observed that the nano-TiO_2_ was dispersed more homogeneously within the PLA matrix than in MB. In addition, the FTIR and DMA analyses allowed the establishment of the presence of interactions between the materials and the TiO_2_ nanoparticles, where the PLA E´value was higher than MB. Regarding ethylene removal, it can be concluded that the samples prepared with PLA at 5% nano-TiO_2_ and with UV light irradiation were significantly more efficient than the other samples, reaching an ethylene removal rate of approximately 68%. This behavior is supported by the better dispersion of the active component in the PLA, while in MB, the agglomeration of the active component was evident. On the other hand, a better effect on ethylene removal was determined using UV light irradiation with respect to VIS, and this is because the photocatalytic activity is more effective under this type of radiation.

As for the in vivo test on bananas, the 5% nano-TiO_2_ PLA samples were also the ones that delayed the ripening of bananas at the end of the 10 days of evaluation, which was significantly reflected in the parameters of color, soluble solids, texture, starch index, and ripening index. Furthermore, it can be concluded that the addition of TiO_2_ has a significant effect on ethylene removal, and therefore, it is a good alternative for improving the shelf life of fresh fruits and vegetables.

## Figures and Tables

**Figure 1 polymers-16-00853-f001:**
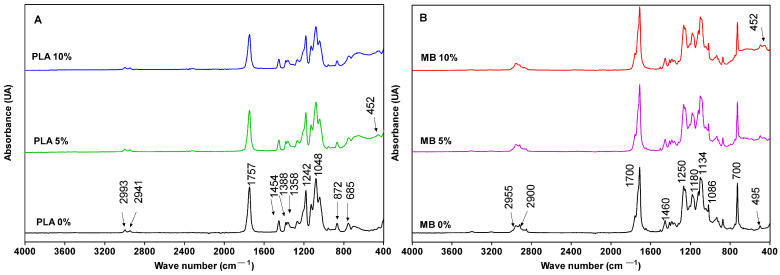
FTIR spectra of PLA (**A**) and MB (**B**) with different concentrations of nano-TiO_2_.

**Figure 2 polymers-16-00853-f002:**
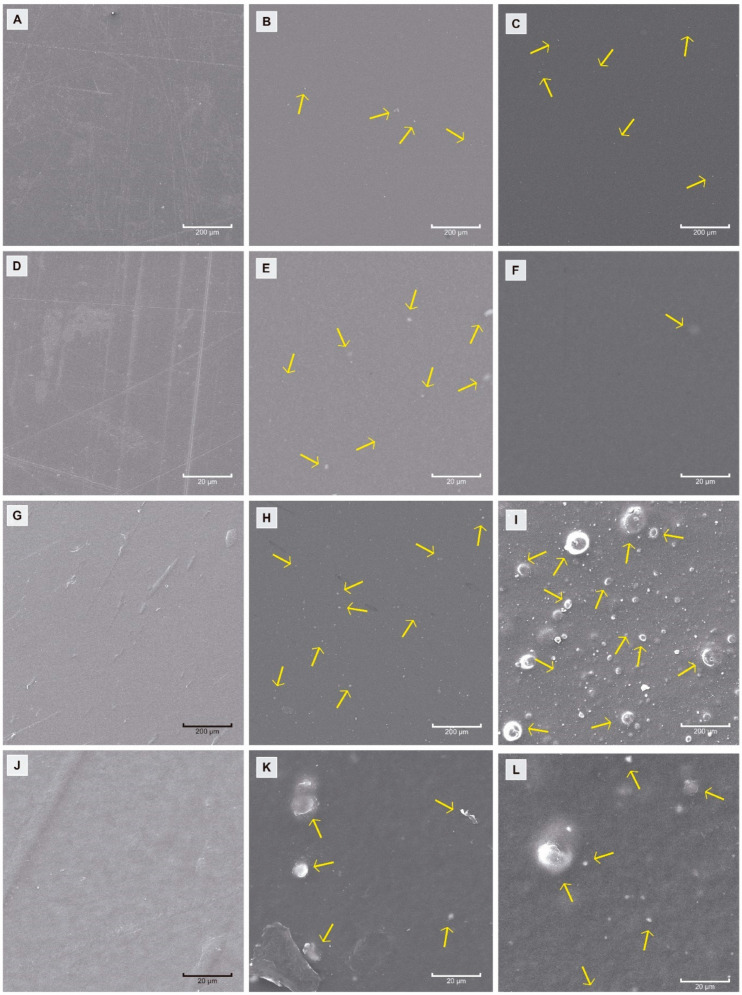
SEM micrographs of active films: PLA 200× 0%, 5% and 10% (**A**–**C**); PLA 1.5 K× 0%, 5% and 10% (**D**–**F**); MB 200× 0%, 5% and 10% (**G**–**I**); and MB 1.5 K× 0%, 5% and 10% (**J**–**L**). TiO_2_ is marked with yellow arrows.

**Figure 3 polymers-16-00853-f003:**
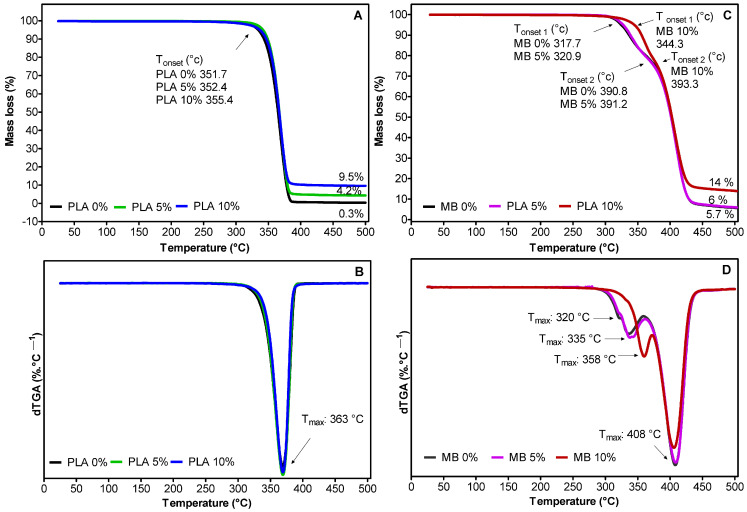
TGA and dTGA thermograms of the active films prepared from PLA (**A**,**B**) and MB (**C**,**D**) with different contents of nano-TiO_2_, respectively.

**Figure 4 polymers-16-00853-f004:**
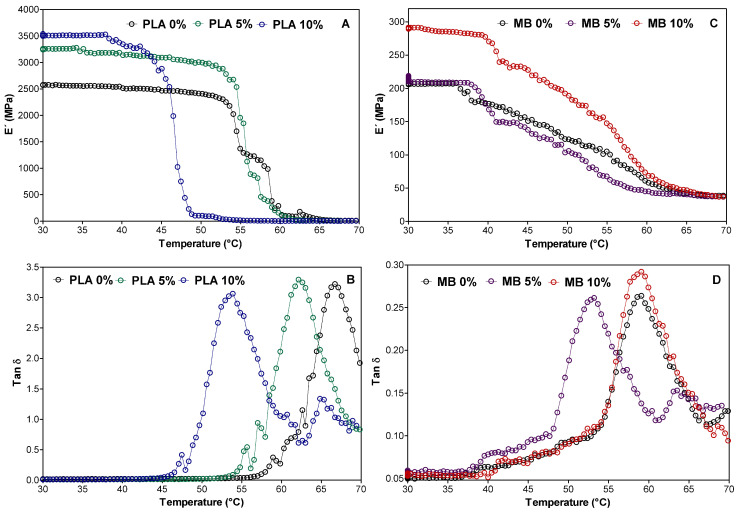
Storage moduli and damping factors (tan δ) of films with PLA (**A**,**B**) and MB (**C**,**D**), respectively.

**Figure 5 polymers-16-00853-f005:**
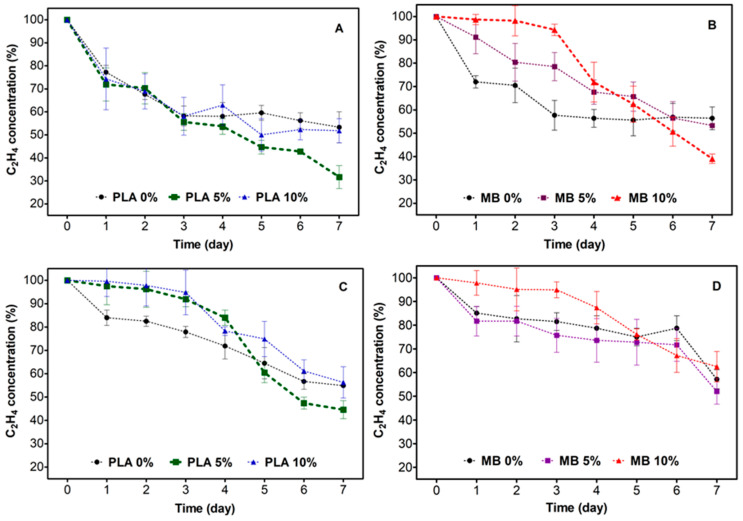
Evolution of ethylene concentration during 7 days for the active materials based on PLA/nano-TiO_2_ and MB/nano-TiO_2_ nanocomposites for UV light irradiation (**A**,**B**) and VIS light irradiation (**C**,**D**).

**Figure 6 polymers-16-00853-f006:**
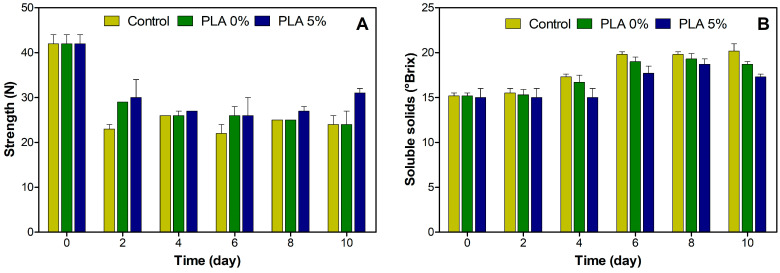
Evaluation of texture (**A**) and soluble solids (**B**) of bananas during storage days.

**Figure 7 polymers-16-00853-f007:**
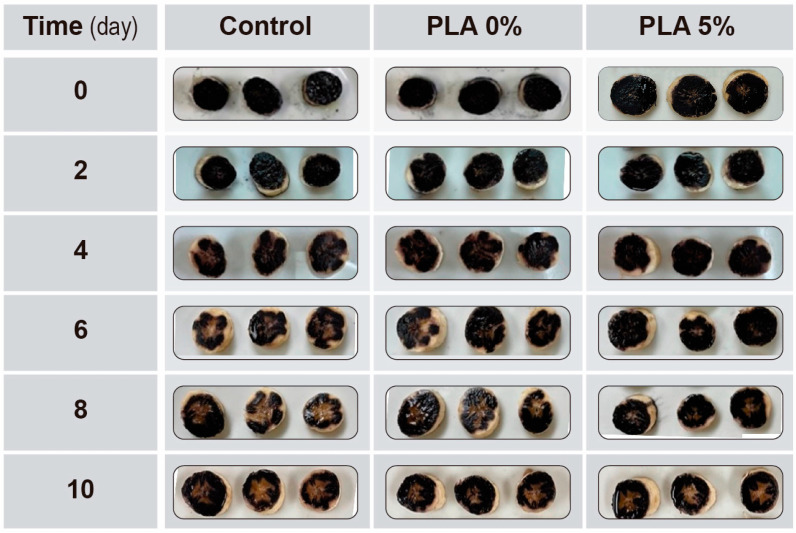
Starch index for bananas wrapped in PLA active films.

**Table 1 polymers-16-00853-t001:** DSC results of PLA film with nano-TiO_2_.

Film	Tg (°C)	Tcc (°C)	ΔHcc (J·g^−1^)	Tm_1_ (°C)	∆Hm_1_ (J·g^−1^)	X (%)
PLA 0%	58.6 ± 0.1 ^A^	128.8 ± 0.9 ^A^	6.4 ± 0.3 ^B^	150.8 ± 0.5 ^A^	7.9 ± 0.1 ^B^	1.6 ± 0.2 ^A^
PLA 5%	58.7 ± 0.1 ^A^	118.3 ± 0.1 ^B^	18.1 ± 1.1 ^A^	148.2 ± 0.2 ^B^	20.2 ± 0.9 ^A^	2.3 ± 0.2 ^A^
PLA 10%	58.4 ± 0.3 ^A^	126.2 ± 1.7 ^A^	6.2 ± 0.9 ^B^	150.2 ± 0.6 ^A^	6.9 ± 0.9 ^B^	0.7 ± 0.1 ^B^

Note: Different letters mean that there are statistically significant differences (*p* < 0.05).

**Table 2 polymers-16-00853-t002:** DSC results of MB film with nano-TiO_2_.

Film	Tg (°C)	Tm_1_ (°C)	∆Hm_1_ (J·g^−1^)	Tm_2_ (°C)	∆Hm_2_ (J·g^−1^)
MB 0%	63.2 ± 0.1 ^A^	122.7 ± 0.1 ^A^	4.6 ± 0.1 ^A^	166.2 ± 0.1 ^A^	0.8 ± 0.2 ^A^
MB 5%	63.1 ± 0.1 ^A^	123.1 ± 0.2 ^A^	4.7 ± 0.3 ^A^	166.1 ± 0.1 ^A^	1.0 ± 0.1 ^A^
MB 10%	63.0 ± 0.2 ^A^	127.1 ± 0.3 ^A^	4.6 ± 0.3 ^A^	166.1 ± 0.2 ^A^	1.3 ± 0.2 ^A^

Note: Different letters mean that there are statistically significant differences (*p* < 0.05).

**Table 3 polymers-16-00853-t003:** Mechanical properties of PLA and MB films with different nano-TiO_2_ concentrations.

Film	TiO_2_ Content (%)	Thickness (µm)	Young’s Modulus (MPa)	Tensile Strength (MPa)	Elongation at Break (%)
PLA	0	65.6 ± 2.2 ^Ba^	2697.3 ± 256.9 ^Aa^	50.5 ± 4.4 ^Aa^	2.2 ± 0.1 ^Bb^
	5	46.1 ± 4.9 ^Bb^	3073.4 ± 199.1 ^Aa^	52.6 ± 3.5 ^Aa^	2.1 ± 0.1 ^Ba^
	10	14.1 ± 0.8 ^Bc^	1247.5 ± 71.7 ^Ab^	16.6 ± 1.6 ^Ab^	1.8 ± 0.1 ^Bc^
MB	0	81.1 ± 1.8 ^Aa^	375.3 ± 12.3 ^Ba^	19.0 ± 3.7 ^Ba^	546.9 ± 84.3 ^Ab^
	5	86.2 ± 7.6 ^Ab^	114.5 ± 3.2 ^Ba^	16.1 ± 0.7 ^Ba^	744.1 ± 19.4 ^Aa^
	10	82.7 ± 4.9 ^Ac^	95.2 ± 4.7 ^Bb^	6.3 ± 0.7 ^Bb^	439.7 ± 43.7 ^Ac^

Note: Different letters mean that there are statistically significant differences (*p* < 0.05); upper case letters: film factor; lower case letters: TiO_2_ content.

**Table 4 polymers-16-00853-t004:** Final ethylene degradation of PLA and MB films.

Matrix	TiO_2_ Content (%)	Final C_2_H_4_ Removal (%)	
		UV	VIS
PLA	0	48.2 ± 5.3	45.1 ± 0.8
	5	68.3 ± 5.0	55.4 ± 3.9
	10	46.7 ± 6.7	43.7 ± 6.7
MB	0	43.6 ± 4.9	42.8 ± 4.0
	5	46.7 ± 0.8	48.0 ± 5.3
	10	60.9 ± 2.0	37.5 ± 6.4

**Table 5 polymers-16-00853-t005:** Color parameters of banana peel during the evaluation days.

Parm.	Film	Time (Day)					
		0	2	4	6	8	10
	Control	67.4 ± 1.9 ^Aa^	70.2 ± 1.7 ^Aa^	65.2 ± 0.3 ^Ab^	66.0 ± 1.0 ^Aab^	64.3 ± 1.9 ^Ab^	64.2 ± 1.0 ^Ab^
	PLA 0%	67.4 ± 1.9 ^Aa^	66.2 ± 1.2 ^Aa^	65.3 ± 0.5 ^Ab^	67.0 ± 1.6 ^Aab^	66.7 ± 0.8 ^Ab^	66.2 ± 1.4 ^Ab^
	PLA 5%	67.4 ± 1.9 ^Aa^	69.0 ± 1.3 ^Aa^	67.1 ± 0.6 ^Ab^	66.8 ± 0.7 ^Aab^	66.7 ± 0.8 ^Ab^	65.5 ± 0.7 ^Ab^
a*	Control	−10.7 ± 2.8 ^Aa^	−5.8 ± 0.9 ^Ab^	−2.7 ± 1.1 ^Ac^	1.1 ± 0.5 ^Ad^	1.2 ± 0.8 ^Ad^	3.2 ± 0.7 ^Ad^
	PLA 0%	−10.7 ± 2.8 ^Ba^	−10.4 ± 1.4 ^Bb^	−6.8 ± 0.4 ^Bc^	−1.2 ± 0.5 ^Bd^	0.8 ± 1.6 ^Bd^	0.0 ± 0.6 ^Bd^
	PLA 5%	−10.7 ± 2.8 ^Ba^	−10.0 ± 1.9 ^Bb^	−6.3 ± 1.7 ^Bc^	−1.0 ± 1.6 ^Bd^	−0.7 ± 1.0 ^Cd^	−0.2 ± 0.4 ^Bd^
b*	Control	43.8 ± 3.0 ^Aa^	42.2 ± 3.3 ^Ab^	35.7 ± 1.7 ^Ac^	33.1 ± 0.6 ^Ad^	34.6 ± 0.8 ^Ac^	39.6 ± 1.2 ^Ad^
	PLA 0%	43.8 ± 3.0 ^Ba^	40.2 ± 0.4 ^Bb^	36.2 ± 0.8 ^Bc^	34.0 ± 1.1 ^Bd^	35.8 ± 2.2 ^Bc^	34.6 ± 2.1 ^Bd^
	PLA 5%	43.8 ± 3.0 ^Ba^	41.0 ± 1.1 ^Bb^	37.9 ± 1.1 ^Bc^	33.6 ± 1.0 ^Bd^	36.8 ± 0.9 ^Bc^	31.6 ± 1.2 ^Bd^
∆E	PLA 0%	-	27.3 ± 1.1 ^Ad^	19.3 ± 0.1 ^Ac^	10.3 ± 1.7 ^Ab^	8.0 ± 1.1 ^Aa^	6.2 ± 1.7 ^Ac^
	PLA 5%	-	17.9 ± 1.5 ^Bd^	6.9 ± 0.1 ^Bc^	9.8 ± 1.0 ^Bb^	3.9 ± 0.6 ^Ba^	2.0 ± 0.3 ^Bc^

Note: Upper case letters indicate that there are significant differences (*p* > 0.05) between the “sample” parameter, while lower case letters indicate that there are significant differences between the “time” parameter.

## Data Availability

Data are contained within the article.

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
