# Peer review of "Ethylene Scavenging Films Based on Ecofriendly Plastic Materials and Nano-TiO2: Preparation, Characterization, and In Vivo Evaluation"

_polymers, 2024, doi:10.3390/polym16060853_

Round 1

Reviewer 1 Report

Comments and Suggestions for Authors

This paper presents the study about development of films based in two types of biopolymers: Mater-Bi (MB) and poly-lactic acid (PLA) with nanoparticles of TiO2 and to determine their ethylene remover capacity and its application in banana. Due to topic, the current study is on the top of relevance and general interest to the readers of the journal. This paper deals with very current problems and could be of scientific and applicative interest of readers.

The paper is well thought out and conceived, and clearly written. The theoretical part is in accordance with the problem. The results are clearly presented, and the methodology is appropriate.

It is necessary to correct technical errors that exist in the paper and, also it is necessary to correct and uniform the literature, in the part References. several references have not been bolded for years. The name of the cited journal - sometimes the full name is given and sometimes the abbreviation. It is necessary to uniform the cited journal according to Instruction.

Page 1, line 19: “dioxide titanium“ should be rewrite as „titanium dioxide”, as it is in the whole paper.

Page 4, line 187: there are some writing mistakes “mL vial with. T 100 μL of destilled water inside . Then,…”.

Page 5, line 209: there is some writing mistake “Samples were taken from the chamber to characterize the fruits. . Analysis included”

Author Response

We thank Reviewer 1 for his/her valuable comments and for accepting our manuscript after minor changes. The answers to the reviews can be found in the attached uploaded file. 

Reviewer 2 Report

Comments and Suggestions for Authors

The paper is well written and very interested. However, the author should discuss better why the PLA produced with 10% of nanoparticles showed a such reduction of thickness and how to avoid it. 

Author Response

We thank Reviewer 2 for his/her valuable comments and for accepting our manuscript after minor changes. The answers to the reviews can be found in the attached uploaded file. 

Reviewer 3 Report

Comments and Suggestions for Authors

In this work, films of PLA and Mater-Bi have been prepared including TiO2 nanoparticles. The morphological, thermal, mechanical, dynamic, barrier and optical properties were evaluated. Also, the ethylene removal was determined and the material was tested with bananas for 10 days.

In my opinion, this study can be improved; the films are not obtained properly, the nanoparticles are not homogeneously dispersed and the data lack coherence. Data discussion could be improved. The conclusions are not supported by the data.

The English and the article in general should be reviewed. There are several mistakes in the manuscript that need to be corrected:

Line 15, what do you refer to with number 5?

Line 87; please, indicate than RH refers to “relative humidity”

Line 153; PA or PLA?

Legend to Figure 3; indicate that C and D are the thermograms of MB

Line 424; please, indicate that Figure 5A refers to UV light and 5B to VIS light.

Line 435; delete “and 5B”

Reference 3 format should be reviewed.

References 20 and 21 are the same.

I hope that the following considerations will help the authors improve the article:

Line 89; if there is no significant difference in the ethylene removal capacity between films with 1 and 2 wt.% TiO2 due to the TiO2 agglomeration over 1 wt.%, why are films with 5 and 10% TiO2 prepared?

Line 115, what is MB (Mater-Bi)? Please, describe the material and indicate related references. Why is this material eco-friendly?

Line 148; the 8 mg samples seem like too much for DSC, is that correct?

Lines 150-152; maybe some more details are needed, I am not sure that it is understandable.

Figure 2 is of poor quality, and the micrographs reveal that the nanoparticles are not dispersed homogeneously. The material is not well prepared.

Lines 285-287; sorry, but the evidence of nanoparticle agglomeration does not indicate that isolated nanoparticles also exist.

Lines 290-293; what do you mean with the expression “all these changes in thermal properties”? What changes? If the addition of nanoparticles does not seem to cause movements in the polymer chains, it cannot be concluded that there are interactions between PLA and the surface of the nanoparticles.

Lines 294-302; what do you mean by these explanations? Is MB made up of PLA, PBAT and TPS? Please, explain it better. Lines 406-412 should be before in the manuscript, in the Introduction for example.

Lines 435-437; what is the reason for the higher and faster removal activity of PLA if the nanoparticle content is the same?

Lines 448-454; why do you develop the experiment with VIS light if you previously know that C2H4 removal is lower than with UV light?

Lines 455-461; the greater interaction of TiO2 with PLA has not been evidenced; the higher ethylene removal with 5% TiO2 than 10% TiO2 is a result of a higher homogeneous distribution, so a better distribution of nanoparticles is needed in the films with 10% TiO2. I think the problem is in the methodology of preparing the films.

Comments on the Quality of English Language

The English and the article in general should be reviewed. There are several mistakes in the manuscript that need to be corrected:

Lines 74-78, redundancy; just one sentence would be enough.

Lines 150-152; I am not sure that it is understandable.

Line 175;  “a universal testing machine” or “an universal testing machine?

Lines 187, 209, 259, 325; revise the points “.” in the sentence.

Line 243; “cans”? What do you mean?

Hyphenation should be reviewed.

Line 355-356; “in the” is repeated.

Line 493, please, review it.

Lines 514-517 and lines 371-378 need revision.

Author Response

We thank Reviewer 3 for his/her valuable comments and for accepting our manuscript after minor changes. The answers to the reviews can be found in the attached uploaded file. 

Reviewer 4 Report

Comments and Suggestions for Authors

The paper describes the development of the biodegradable films with TiO2 nanoparticles for the photo-assisted removal of ethylene from fruit packages. The manufactured materials were characterized by FTIR,  SEM, thermal analysis (DSC and TGA), FTIR give some information about the starting polymer but very little about both composite materials and this spectral part is not discussed. Normally concentration of TiO2 should have impact on spectra in 800-400 cm-1 range.  Also I would recommend to take larger resolution for SEM. EDX could also better support incorporation and mapping of TiO2 particles.

 In the present form Figure 3 C provides the same mass loss by heating up to 500 C for with different TiO2 content. PLA? Or MB? samples is not clear. Definitely wrong description and result itself should be checked.

Results definitely have a value but the presentation of research should be clarified and improved.

Author Response

We thank Reviewer 4 for his/her valuable comments and for accepting our manuscript after minor changes. The revisions are detailed in the attached file
